# The Role of the Beclin1 Complex in Rab9-Dependent Alternative Autophagy

**DOI:** 10.3390/ijms26189151

**Published:** 2025-09-19

**Authors:** Sohyeon Baek, Yunha Jo, Jihoon Nah

**Affiliations:** 1Department of Biological Sciences and Biotechnology, Chungbuk National University, Cheongju 28644, Republic of Korea; 2Department of Biochemistry, Chungbuk National University, Cheongju 28644, Republic of Korea

**Keywords:** Beclin1 complex, alternative autophagy, Rab9, TMEM9, WD-repeat protein, phosphoinositide interacting (Wipis), heart diseases

## Abstract

Autophagy is a conserved catabolic pathway that degrades intracellular cargo through the lysosomal system. Canonically, this process is orchestrated by the autophagy-related (Atg)5-Atg7 conjugation system, which facilitates the formation of microtubule-associated protein 1 light chain 3 (LC3)-decorated double-membrane vesicles known as autophagosomes. However, accumulating evidence has revealed the existence of an Atg5-Atg7-independent, alternative autophagy pathway that still relies on upstream regulators such as the unc-51 like autophagy activating kinase 1 (Ulk1) kinase and the Beclin1 complex. In this review, we provide a comprehensive overview of the role of the Beclin1 complex in canonical autophagy and highlight its emerging importance in alternative autophagy. Notably, the recent identification of transmembrane protein 9 (TMEM9) as a lysosomal protein that interacts with Beclin1 to promote member RAS oncogene family 9 (Rab9)-dependent autophagosome formation has significantly advanced our understanding of alternative autophagy regulation. Furthermore, this Ulk1-Rab9-Beclin1-dependent mitophagy has been shown to mediate to mitochondrial quality control in the heart, thereby contributing to cardioprotection under ischemic and metabolic stress conditions. We further examine how the Beclin1 complex functions as a central scaffold in both canonical and alternative autophagy, with a focus on its modulation by novel factors such as TMEM9 and the potential therapeutic implications of these regulatory mechanisms.

## 1. Introduction

Autophagy is a fundamental cellular homeostasis mechanism by which cells degrade and recycle intracellular proteins or organelles through lysosomal digestion. The widely characterized form of autophagy proceeds through the formation of double-membraned autophagosomes that sequester bulk cytosolic materials and deliver them to lysosomes for degradation. In this canonical autophagy, autophagosome generation is orchestrated by dedicated Atg proteins such as Ulk1, Atg5, Atg7 and LC3 [1,2,3,4,5]. In nutrient-rich conditions, the Ser/Thr kinase mammalian target of rapamycin complex 1 (mTORC1) keeps autophagy repressed through the suppressive phosphorylation on Ulk1 kinase at Ser638 and Ser758 [6]. Upon nutrient-poor or stress conditions, mTORC1 inhibition and protein kinase AMP-activated catalytic subunit alpha 1 (AMPK) activation promote the activation of the Ulk1 kinase complex by enhancing phosphorylation of Ulk1 at Ser317, Ser637 and Ser777, thereby facilitating its translocation to nascent autophagic membranes [7,8]. Ulk1 then activates the class III phosphatidylinositol 3-kinase (PI3K) complex, often termed the Beclin1 complex, at the phagophore formation site [9]. This triggers local generation of phosphatidylinositol-3-phosphate (PI3P), recruiting downstream effectors that expand the isolation membrane into an autophagosome [10]. The subsequent elongation steps involve two ubiquitin-like conjugation cascades, the Atg7-Atg3-mediated lipidation of LC3 and the formation of the Atg5-Atg12-Atg16L1 complex, both of which contribute to decorating autophagosome membranes with LC3-PE (LC3-II). This canonical pathway is characterized by LC3 lipidation and requires Atg5 and Atg7, which were initially considered as essential for autophagosome formation [11,12].

It was paradigm-shifting when the Dr. Shimizu group discovered an alternative autophagy pathway that enables autophagosome formation independently of Atg5 and Atg7. In Atg5- or Atg7-deficient cells, autophagosomes and autolysosomes can still form without LC3 lipidation (LC3-II), and this phenomenon was observed using transmission electron microscopy (TEM) [5]. This Atg5-Atg7-independent autophagy (also termed “alternative autophagy”) has challenged the long-standing notion that LC3 conjugation systems are absolutely required for autophagosome generation. Subsequent studies confirmed that this noncanonical pathway is a bona fide autophagic process rather than an artifactual observation [13,14,15]. Importantly, although alternative autophagy does not require certain essential Atg genes, such as Atg5 and Atg7, it shares some upstream machinery with canonical autophagy, including Ulk1 and the Beclin1 PI3K complex [5,16]. The critical divergence between canonical and alternative autophagy lies in the membrane source and mechanism of autophagosome formation. Instead of employing LC3 for membrane elongation and closure, the alternative pathway relies on Rab9-dependent utilization of endosomal and Golgi-derived membranes to form autophagosomes. As will be discussed in detail, autophagosomes in alternative autophagy appear to originate from the trans-Golgi network (TGN) and late endosomes, through a process mediated by the small GTPase Rab9 [5]. Consequently, alternative autophagosomes are largely LC3-negative but Rab9-positive, reflecting their unique biogenesis route [16].

This review will first provide a brief overview of canonical autophagy machinery with an emphasis on the Beclin1 complex and its interactomes. We will then delve into the molecular mechanism of Rab9-dependent alternative autophagy, summarizing key discoveries that have illuminated how cells assemble autophagosomes without the Atg5-Atg7-LC3 conjugation system. A major focus will be the function of the Beclin1 complex in this alternative context and how it is recruited and activated by newly identified regulators. In particular, we highlight recent insights into TMEM9, a lysosomal transmembrane protein that interacts with Beclin1 to trigger Rab9-mediated autophagosome formation [17]. Finally, we consider the physiological relevance of alternative autophagy in mammals and its potential implications in disease, with a particular focus on cardiac conditions. Interestingly, the formation of Rab9-positive puncta upon glucose deprivation is markedly suppressed in Ulk1- or Beclin1-deficient cells, indicating that Rab9 recruitment to autophagosomes and autolysosomes relies on Ulk1- and Beclin1-mediated autophagic machinery [5,13]. Moreover, recent studies have demonstrated that Rab9-dependent mitophagy confers a cardioprotective effect in pathological contexts such as cardiac pressure overload and ischemia, further highlighting the physiological significance of this noncanonical pathway [13,14]. Understanding the dual involvement of the Beclin1 complex in both canonical and alternative autophagy provides a more complete picture of autophagic regulation and offers new perspectives for therapeutic intervention in diseases where autophagy is dysregulated.

## 2. The Beclin1 Complex in Canonical Autophagy

Beclin1 was the first characterized mammalian autophagy gene, identified by Dr. Levine’s group, originally recognized through its binding to the apoptosis regulator Bcl-2 [18]. It is now widely known as a core component of the class III PI3K complex that drives autophagosome nucleation. Beclin1 is the mammalian ortholog of yeast Atg6/Vps30, highlighting its evolutionarily conserved role in autophagy [19]. In canonical autophagy, Beclin1 serves as a scaffold for assembly of the core nucleation complex, which consists of the lipid kinase Vps34 (PIK3C3) and its regulatory subunit Vps15 (PIK3R4; p150) [20]. This core complex interacts with various accessory subunits that determine the functional destination of this complex. The major autophagy-enhancing factor of the Beclin1 complex is Atg14L, which is recruited to the nascent phagophore to generate PI3P and initiate autophagosome formation [21]. In other contexts, Beclin1 forms complexes with alternative partners such as UV radiation resistance-associated gene protein (UVRAG) or run domain Beclin1-interacting and cysteine-rich domain-containing protein (Rubicon), thereby forming distinct complexes that regulate endosomal maturation, autophagosome-lysosome fusion, or suppression of autophagy [21,22]. Thus, Beclin1 acts as a nexus integrating signals that regulate autophagy initiation, vesicle trafficking, and crosstalk with other vesicular pathways (Table 1 and Figure 1 and Figure 2).

A critical aspect of Beclin1 regulation involves its interaction with Bcl-2 family proteins. Beclin1 contains a Bcl-2 homology 3 (BH3) domain through which it can bind the anti-apoptotic proteins Bcl-2 or Bcl-X_L_ (Bcl2L1) [23,24]. In growth conditions, Bcl-2 sequesters Beclin1 at ER membranes, preventing unwarranted autophagy [24]. Under stress, pro-apoptotic BH3-only proteins or stress-activated c-Jun N-terminal kinase (JNK) phosphorylation of Bcl-2 can disrupt the Beclin1-Bcl-2 interaction, releasing Beclin1 to assemble the active Beclin1 complex [30]. This release of Beclin1 is a key triggering event for autophagy initiation. Notably, autophagy-inducing peptides such as TAT-Beclin1 have been developed to mimic this release mechanism, promoting autophagy by enhancing Beclin1 activity independently of upstream signals. TAT-Beclin1 binds to GAPR1 (also known as GLIPR2), a negative regulator of autophagy that interacts with Beclin1 at the Golgi, thereby displacing it and further promoting autophagosome formation [27]. These peptides provide promising therapeutic interventions in conditions where autophagy is impaired.

Numerous other regulatory proteins interact with Beclin1 to modulate its activity and localization. For example, AMBRA1, Atg14L, and Bif-1 promote autophagy by facilitating Beclin1 complex formation, whereas Rubicon is inhibitory [25,26,28,31]. Beclin1 forms a complex with AMBRA1, which, under nutrient-rich conditions, binds to LC1 to anchor the complex to microtubules. Under starvation conditions, AMBRA1 is phosphorylated by Ulk1, leading to a dissociation of the AMBRA1-Beclin1 complex from DLC1. This phosphorylation event facilitates the translocation of the AMBRA1-Beclin1-Vps34 complex to the ER, where autophagosome nucleation is initiated [25,32]. In addition, Bif-1 interacts with Beclin1 indirectly through UVRAG, and its absence suppresses PI3KC3 complex activity under starvation conditions, thereby promoting tumorigenesis [28]. By contrast, Rubicon forms a stable complex with UVRAG, Beclin1, Vps34, and Vps15, but not with Atg14L. Loss of Rubicon enhances LC3 puncta formation and promotes both autophagosome maturation and endocytic trafficking [21]. Collectively, these mechanisms ensure that the Beclin1 complex is activated both temporally and spatially, coordinating early autophagy signals with downstream membrane remodeling and autophagosome formation.

Post-translational modifications (PTMs) of Beclin1 also influence its function. In particular, multi-site phosphorylation by kinases such as Ulk1, death associated protein kinase 1 (DAPK), AMPK, Akt and mammalian Ste20-like kinases 1 (Mst1) can modulate its interactions with binding partners and either enhance or inhibit Vps34 lipid kinase activity, thereby fine-tuning the initiation of autophagy [33,34,35,36]. Phosphorylation of Beclin1 at Ser14 by Ulk1 is required for the activation of the Vps34 complex kinase activity upon amino acid starvation or mTOR inhibition [35]. DAPK phosphorylates Beclin1 at Thr119 within its BH3 domain, the region responsible for interacting with Bcl-2 family proteins. This phosphorylation disrupts the association between Beclin1 and the anti-apoptotic protein Bcl-X_L_ thereby promoting autophagy [33]. In addition, AMPK promotes autophagy by phosphorylating Beclin1 at Ser91 and Ser94, as well as Vps34 at Thr136 and Ser165 [34]. Akt associates Beclin1 and phosphorylates it at Ser234 and Ser295, leading to the suppression of autophagy [36]. Mst1 phosphorylates Beclin1 at Thr108 within its BH3 domain, enhancing its interaction with Bcl-2 or Bcl-X_L_ and suppressing autophagy [37]. Additionally, a recent study identified S-palmitoylation of Beclin1 by the palmitoyl transferase zinc finger DHHC5 as essential for efficient autophagy. Loss of this lipid modification contributes to the age-associated decline in autophagic activity [29]. These upstream modifications not only activate the Beclin1 core complex but also influence downstream events such as PI3P production, phagophore nucleation, and membrane recruitment of downstream effectors like Wipi proteins and Atg complexes (Table 2).

The generation of PI3P by the Beclin1 complex is a pivotal early event in autophagy initiation. PI3P serves as a membrane lipid signal that recruits downstream effector proteins to the nascent phagophore, orchestrating the spatial assembly of the autophagy machinery [10]. Wipi proteins, composed of seven WD40 repeats forming a β-propeller structure, act as PI3P effectors critical for autophagy initiation [38]. Among them, Wipi1 and Wipi2 are key mediators in early phagophore formation in canonical autophagy. Wipi1 binds PI3P to facilitate membrane recruitment, supporting Wipi2 function, while Wipi2 directly recruits the Atg12-Atg5-Atg16L1 complex and promotes LC3 lipidation, essential for autophagosome formation [39]. In contrast, Wipi3 and Wipi4 participate in autophagosome maturation and phagophore expansion by interacting with Atg2 and regulatory kinase complexes, including the AMPK-Ulk1 signaling axis and the tuberous sclerosis complex (TSC)-mTORC1 pathway [40]. Detailed mechanisms underlying these interactions and their functional implications will be discussed comprehensively in the subsequent sections. Double FYVE domain-containing protein 1 (DFCP1), another PI3P-binding protein, localizes to PI3P-enriched omegasomes on the ER and is thought to mark the sites of autophagosome formation [10]. Although both proteins recognize PI3P, DFCP1 specifically initiates the formation of omegasomes, to which Wipi2 is subsequently recruited, acting downstream to promote phagophore elongation and LC3 conjugation [39,41]. Knockdown of Wipi2 in human embryonic kidney 293A (HEK293A) cells results in accumulation of DFCP1-positive structures, and this accumulation persists under starvation conditions, indicating that Wipi2 is essential for proper omegasome maturation and progression into functional autophagosomes [41]. Together, these PI3P effectors form dynamic scaffolds that regulate membrane curvature, elongation, and maturation of the autophagosome, underscoring the critical role of Beclin1-mediated PI3P production in coordinating downstream autophagy events.

In summary, in canonical Atg5-Atg7-dependent autophagy, the Beclin1 complex is essential for initiating isolation membrane nucleation by generating PI3P [42]. This lipid signal recruits PI3P-binding effectors such as Wipi proteins and DFCP1, which subsequently facilitate the recruitment of the Atg5-Atg12-Atg16L1 complex and LC3, thereby promoting membrane expansion and autophagosome closure. The classical function of Beclin1 in this process is tightly controlled by nutrient and stress signals through its many interactors [43]. Understanding these detailed molecular mechanisms helps explain how the Beclin1 complex can also drive an alternative form of autophagy that does not require the Atg5-Atg7 system, yet still produces fully functional autophagosomes.

## 3. Rab9-Dependent Alternative Autophagy: Overview and Detailed Mechanism

Alternative autophagy refers to a form of macroautophagy that proceeds without the Atg5 and Atg7-driven ubiquitin-like LC3 conjugation machinery. The existence of this pathway was first demonstrated by Dr. Shimizu’s group, who showed that Atg5^−/−^ or Atg7^−/−^ mouse embryonic fibroblasts (MEFs) can form double-membraned autophagosomes and degrade substrates through lysosome under certain stress conditions, as revealed by transmission electron microscopy [5]. Hallmarks of this process included the absence of LC3-decorated autophagosomes, even though autophagic vesicles clearly formed and fused with lysosomes. In addition to its novel findings, this study demonstrated that key autophagy-related proteins such as Ulk1 and Beclin1 are essential for the alternative autophagy pathway, mirroring their roles in canonical autophagy [5]. These findings support a model in which the early stages, upstream signaling and phagophore nucleation via PI3K, might be common to both canonical and alternative autophagy, with divergence occurring during membrane expansion and autophagosome completion. However, an alternative possibility is that these two autophagy pathways are differentially regulated through context-dependent reconstruction of the Beclin1 interactome, which may confer distinct functional outputs despite shared initiation steps (Table 3 and Figure 3). This possibility will be discussed in more detail later in the review.

One of the most striking differences between the canonical and the alternative pathway is the membrane origin of the autophagosomes. Unlike canonical autophagy, which primarily utilizes ER-derived membranes, alternative autophagy utilizes membranes originating from the TGN and late endosomes [5]. The small GTPase Rab9, which generally involved in late endosome-to-TGN transport, plays an essential role in this alternative process. A previous study demonstrated that alternative autophagosome formation is dependent on Rab9, and that silencing Rab9 effectively blocked autophagosome generation in Atg5-deficient cells [5,49]. From a mechanistic perspective, Rab9 is thought to facilitate the delivery or tethering of Golgi/endosomal membrane fragments to the growing isolation membrane, compensating for the absence of LC3-mediated membrane elongation [5]. Indeed, live-cell imaging and colocalization studies have demonstrated the presence of GFP-Rab9 on autophagosome-like vesicles that subsequently mature into autolysosomes, supporting its direct role in autophagosome formation [13]. In alternative autophagy, autophagosomes formed via this pathway often contain markers of late endosomes or TGN, and can fuse with lysosomes to degrade their cargo, confirming that the pathway is complete and functional. Thus, just as LC3 puncta formation is widely used to monitor autophagic activity in canonical autophagy, the appearance of Rab9-positive puncta and their fusion with lysosomes serves as a key indicator of alternative autophagy [5].

Similarly to canonical autophagy, the initiation of alternative autophagy is mediated by the Ulk1 complex. Ulk1 is activated upon stress and translocates to potential membrane sources to trigger autophagosome formation. However, the mechanism by which Ulk1 is activated in alternative autophagy may differ from the canonical mechanisms. One example is that genotoxic stress, such as DNA damage induced by etoposide, can trigger alternative autophagy in a Ulk1-dependent but mTORC1-independent manner [44]. In this study, phosphorylation of Ulk1 at Ser746 was identified as specifically required for the induction of alternative autophagy under this condition [44,61]. Defective mutation of this site abrogated autophagosome formation in Atg5^−^^/*−*^ cells upon DNA damage, without affecting canonical autophagy. This suggests that upstream kinases, possibly DNA-damage-responsive kinases, selectively modify Ulk1 to initiate the alternative autophagy program. Indeed, p53-induced Ripk3 phosphorylates Ulk1 at Ser746 in response to genotoxic stress, promoting its Golgi translocation and triggering alternative autophagy [44].

Furthermore, in cardiomyocytes under energy stress, AMPK was found to activate Ulk1, which in turn initiates alternative autophagy [13]. Once activated, Ulk1 orchestrates downstream events during alternative autophagy, including the regulation of Rab9. Specifically, Ulk1-mediated phosphorylation of Rab9 at Ser179 has been shown to be essential for its role in autophagosome formation [13,62]. In cardiomyocytes, Ulk1 directly phosphorylates Rab9, that promotes the assembly of a multi-protein complex at the autophagosome formation site [13,62]. This complex, identified in the context of alternative mitochondria-specific autophagy, includes Ulk1, Rab9, receptor-interacting Ripk1 and dynamin-related protein 1 (Drp1). Rab9 phosphorylated by Ulk1 recruits Ripk1, which in turn phosphorylates Drp1 at Ser616. Activated Drp1 mediates mitochondrial fission, segregating damaged mitochondria and facilitating their encapsulation by Golgi-derived autophagosomes [13]. In this manner, a Ulk1-Rab9-Ripk1-Drp1 signaling axis drives the formation of Rab9-positive autophagosomes around mitochondria, independently of the LC3-conjugation system. Notably, genetic evidence underscores the importance of this signaling axis. Knock-in mice expressing a phosphorylation-deficient Rab9 mutant exhibit defective alternative cardiac mitophagy and suffer exacerbated myocardial injury, despite having intact canonical autophagy [13]. Likewise, Ulk1 knockout abrogates alternative autophagosome formation while leaving basal canonical autophagy largely unaffected, due to the compensatory role of Ulk2 [44,57]. These findings establish that Ulk1-dependent, Rab9-mediated autophagy is a distinct pathway vital for cellular adaptation to certain stresses.

Given the absence of the LC3 conjugation system in alternative autophagy, how are the later steps of autophagosome maturation accomplished? In canonical autophagy, LC3 and other Atg8-family proteins, such as GABARAP families decorate the inner and outer autophagosomal membrane, aiding in cargo recruitment and also in membrane closure and fusion, through binding to LC3-interacting region (LIR)-containing adaptor proteins and tethering factors [1,63]. Through their LIR motifs, adaptor proteins such as PLEKHM1 and EPG5 are recruited to LC3/GABARAP-decorated membranes, thereby bridging autophagosomes with lysosomes. PLEKHM1, in particular, binds to both LC3 and Rab7, thereby contributing to the stabilization of HOPS-mediated membrane tethering prior to fusion. EPG5 interacts with LC3 and the SNARE complex STX17- SNAP29, thereby positioning the autophagosomal membrane for fusion with the lysosomal SNARE protein VAMP8 [54,55]. Given the lack of LC3 involvement, alternative autophagy may rely on noncanonical factors to fulfill these roles, but certain canonical components may still be shared between the two pathways. Researchers in recent years have begun to identify such factors, providing insight into the membrane dynamics of the alternative pathway.

### 3.1. Wipi3/4 in Alternative Autophagy

A breakthrough was achieved by Dr. Shimizu’s group, who identified Wipi3 as an essential factor for alternative autophagy [50]. As aforementioned, Wipi proteins bind PI3P and act as effector scaffolds during autophagy [51]. In canonical autophagy, Wipi1/2 help recruit the LC3 lipidation machinery, but Wipi3 was found to play only a minor role in canonical autophagy [39,40]. Instead, Wipi3 is critical in the alternative pathway, where it localizes to Golgi membranes and is required for generating the isolation membrane for alternative autophagy [50]. Neuron-specific Wipi3 knockout mice exhibited severe neurodegeneration, similar to phenotypes seen in Atg5 knockouts, indicating that alternative autophagy sustains neuronal health when canonical autophagy is impaired. Interestingly, Atg5/Wipi3 double knockout mice had exacerbated neurodegenerative defects and were largely embryonic lethal [50]. These genetic interactions underscore that Wipi3-dependent alternative autophagy functions in parallel to canonical autophagy to maintain homeostasis, especially in long-lived cells like neurons. Wipi3 may help recruit membranes or membrane-shaping factors to the growing autophagosome in the absence of LC3. Indeed, the presence of Wipi3 on Golgi/TGN could facilitate the capture of those membranes into the autophagosome. Loss of Wipi3 impairs the formation of alternative autophagosomes, highlighting its specific requirement [50].

Although less well-characterized, Wipi4, also known as WDR45, may also participate in autophagy-related membrane dynamics. Wipi4 binds PI3P and shares structural similarity with Wipi3, raising the possibility of functional redundancy or context-dependent roles. Notably, Wipi4 interacts specifically with Atg2A, forming a Wipi4-Atg2 complex that associates with Ulk1 and AMPKα to constitute a larger protein complex [40]. However, its specific contribution to alternative autophagy remains unclear. Some studies suggest a supportive or compensatory function, but definitive evidence for its involvement in LC3-independent autophagosome formation is currently lacking. Thus, while Wipi3 is firmly established as essential for alternative autophagy, the role of Wipi4 in this pathway warrants further investigation.

### 3.2. Isolation Membrane Closure via Dram1

Another key player is Dram1, a p53-inducible lysosomal membrane protein known to promote autophagy [52]. In the context of DNA damage, Dram1 has been shown to be pivotal for alternative autophagy induction [45]. Nagata et al. found that genotoxic stress triggers p53 to transcriptionally upregulate Dram1, which is both necessary and sufficient to drive alternative autophagy. Dram1 appears to act at the stage of isolation membrane closure. It localizes to Golgi/endosomal-lysosomal membranes, and its overexpression can induce autophagosome formation even when Atg5 is absent [45,64]. Functionally, Dram1 helps complete the sealing of the double-membrane autophagosome, a step normally facilitated by LC3 in canonical autophagy. The requirement for Dram1 is underscored by the fact that expressing Dram1 can rescue alternative autophagy defects [45]. For instance, introducing Dram1 into Wipi3-deficient cells or tissues can restore autophagosome formation and ameliorate neurodegenerative phenotypes [50]. Thus, Dram1 might work in coordination with Ulk1-Rab9 signaling to ensure the autophagosome membrane closes and fuses properly in the absence of the LC3 conjugation system. It has been proposed that Dram1 could influence membrane curvature or recruit membrane-fusion machinery to facilitate autophagosome completion [45]. In summary, p53-Dram1 signaling defines a genotoxic stress-specific pathway for activating alternative autophagy, illustrating how autophagy is differentially regulated depending on the type of cellular stress.

### 3.3. Syntaxin7 in Membrane Closure

Syntaxin7, a SNARE protein primarily associated with late endosome–lysosome fusion, has been suggested as a potential mediator of membrane fusion events in alternative autophagy [5]. Unlike canonical autophagy, which depends on Syntaxin17 for autophagosome–lysosome fusion, alternative autophagy proceeds in the absence of LC3 lipidation, suggesting the involvement of distinct fusion machinery [48]. In this context, Syntaxin7 has been detected on Rab9-positive autophagosome membranes in Atg5-deficient MEF cells, implicating it as a key contributor to lysosomal fusion in the alternative pathway [5]. In principle, Rab9-dependent alternative autophagy constitutes a parallel route to achieve the same endpoint of cargo degradation through the lysosome by employing a partially distinct set of proteins. While it shares core upstream components with the canonical pathway, such as the Ulk1 kinase and the Beclin1 complex, it replaces the downstream machinery with alternative factors including Rab9, Drp1, Wipi3, and Dram1, which mediate membrane delivery, expansion, and fusion. In the next section, we focus on the Beclin1 complex in this alternative context to examine how Beclin1 is differentially recruited and activated during alternative autophagy.

## 4. The Beclin1 Complex in Alternative Autophagy

That the Beclin1 complex is also required for alternative autophagy was firmly established by early studies. In Atg5^−/−^ MEFs, inhibition of Vps34 or knockdown of Beclin1 prevents autophagosome formation in response to stresses that would otherwise induce the alternative pathway [5]. In other words, regardless of LC3 lipidation or Rab9 association, the generation of PI3P by the Beclin1 complex remains essential for the formation of any type of autophagosome [5,43]. This highlights that the role of Beclin1 as a membrane nucleator is fundamental and conserved across both autophagy pathways. However, there are intriguing questions about how the Beclin1 complex is regulated differently during alternative autophagy. Are the composition and localization of the Beclin1 complex the same in canonical autophagy, or do they differ? What proteins or signals recruit it to the Golgi/endosomal membranes instead of ER? These essential aspects remain unclear and require further investigation to be fully elucidated. Some clues have begun to emerge that may help explain these differences. A logical hypothesis was that the Beclin1 complex might partner with a different set of accessory subunits in alternative autophagy. In our recent work, we were the first to demonstrate that TMEM9 serves as a pivotal link between Beclin1 and alternative autophagy at the lysosomal and endosomal interface [17]. TMEM9 is a single pass type I membrane protein predominantly residing in late endosomes and lysosomes [65]. It was previously known to regulate lysosomal acidification via interaction with the v-ATPase and to be involved in Wnt signaling and inflammation [66,67]. Recently, we first discovered a novel role of TMEM9 in alternative autophagy. TMEM9 binds directly to Beclin1 and activates the Beclin1 complex in alternative autophagy, without affecting canonical autophagy. The interaction is mediated by the cytosolic domain of TMEM9 and the BH3 domain of Beclin1, which is the same site where Bcl-2 binds. In fact, TMEM9 binding displaces Bcl-2 from Beclin1, thereby unleashing Beclin1 to cytoplasm for triggering pro-autophagy function [17]. This represents a clever mechanism in which TMEM9 functions as a positive regulator by releasing Beclin1 from its inhibitor, Bcl-2, precisely at the site where alternative autophagosomes form. Consistent with this, co-immunoprecipitation experiments show that TMEM9 and Beclin1 interaction increases under autophagy-inducing conditions, even in the presence of Bcl-2, suggesting that TMEM9 competitively displaces Bcl-2 from Beclin1 during stress. Notably, TMEM9 is localized on late endosomal/lysosomal membranes and was found to partially colocalize with Rab9-positive structures, but not with LC3-positive vesicles. This implies that TMEM9 might recruit Beclin1 complex to Rab9-marked membrane sites, helping nucleate the alternative autophagosome right where it needs to form [17]. Furthermore, TMEM9 requires proper lysosomal targeting to function. It contains three N-linked glycosylation sites in its luminal domain, and these modifications are necessary for its stability and localization in lysosomes [65]. TMEM9 mutants that are not fully glycosylated fail to localize correctly and cannot promote alternative autophagy. Thus, glycosylation-dependent lysosomal localization of TMEM9 is critical for it to engage Beclin1 and activate alternative autophagy at the Rab9-positive vesicles [17].

Other regulatory factors influencing the Beclin1 complex in alternative autophagy include those that modulate availability of Beclin1 or complex composition under specific conditions. For instance, tripartite motif (TRIM) family proteins have attracted growing interest. TRIM31, an E3 ubiquitin ligase primarily expressed in intestinal cells, was found to drive an alternative autophagy pathway crucial for gut immunity [15]. The mechanism of TRIM31 is quite distinct. It directly binds to phosphatidylethanolamine (PE) on membranes via a palmitoylation-dependent lipid-interacting motif. Through this mechanism, TRIM31 can promote the expansion and fusion of isolation membranes, effectively compensating for the lack of LC3-PE in Atg5-Atg7-deficient cells. Notably, TRIM31 is anchored in mitochondrial membranes and is rapidly relocalized or activated upon Toll-like receptor stimulation to induce autophagy in intestinal epithelial cells. In the absence of TRIM31, Atg5^−/−^ or Atg7^−/−^ cells struggle to form autolysosomes and fail to clear intracellular bacteria. TRIM31 thus appears to facilitate the elimination of invasive bacteria through an alternative autophagy pathway. This is particularly significant because mutations in Atg16L1 associated with Crohn’s disease impair canonical autophagy yet still permit autophagy to proceed via this alternative route [15]. While TRIM31 does not directly bind Beclin1, it underscores how other membrane-interacting proteins can assist Beclin1-driven autophagosome formation by catalyzing membrane dynamics such as expansion or fusion. The ability of TRIM31 to bind PE is particularly noteworthy, it suggests TRIM31 might functionally substitute for LC3 in bridging or tethering membranes, thereby promoting autophagosome closure in alternative autophagy [15]. In essence, TRIM31 sets the stage for the Beclin1 complex to complete autophagosome biogenesis in the absence of LC3. Additionally, studies on ultraviolet B (UVB)-induced stress in human keratinocytes provide supporting evidence. While pharmacological inhibition of autophagy markedly amplified NLR family pyrin domain containing 3 (NLRP3) inflammasome activation, genetic ablation of Atg5 or Atg7 failed to suppress IL-1β production. In contrast, knockdown of Beclin1 significantly enhanced inflammasome activity, underscoring that alternative autophagy plays a critical role in the clearance of UVB-induced inflammasome activation [68].

In summary, the Beclin1 complex also occupies a central role in the alternative autophagy pathway, just as it does in canonical autophagy. However, it is subject to a unique regulatory environment in the alternative context. Figure 2 provides a schematic of how various regulators impinge on the Beclin1 complex and the alternative autophagy machinery, highlighting differences from canonical autophagy. The interaction of Beclin1 with Bcl-2 is a common control point. Alternative autophagy employs factors like TMEM9 and possibly BH3-only proteins or Dram1 downstream of p53 to free Beclin1 [17]. The localization of the Beclin1 complex shifts to endosomal or Golgi membranes, facilitated by Rab9 and possibly Wipi3, rather than the ER [5,50]. Furthermore, membrane remodeling and closure tasks are handled by proteins such as Dram1 and TRIM31 instead of the Atg conjugation system [15,45]. Ultimately, the Beclin1 complex serves as a platform for assembling an alternative autophagosome initiation site once properly activated and localized by these factors. In cutaneous disorders, alternative autophagy functions to suppress inflammasome hyperactivation by facilitating the clearance of damaged mitochondria [68]. Given the distinct molecular players involved, it is likely that additional, yet unidentified regulators specific to the alternative autophagy pathway exist. Uncovering these regulators will be crucial for fully understanding the unique mechanisms and physiological roles of this noncanonical autophagic process. In the next section, we will explore the physiological significance of this pathway, with a particular focus on the heart, and discuss the contexts in which cells appear to rely on alternative autophagy rather than the canonical pathway.

## 5. Physiological and Pathophysiological Roles of Alternative Autophagy

Beyond its characterization in experimental settings, the Ulk1-Rab9-dependent alternative autophagy pathway has been found to operate in several physiologically and pathologically relevant situations. Notably, it plays a critical role in mitochondrial clearance during fetal erythropoiesis, where canonical autophagy is dispensable, but Ulk1-dependent alternative autophagy is essential [57]. Likewise, during somatic cell reprogramming, this pathway facilitates the metabolic transition by promoting mitochondrial elimination independently of the canonical machinery [58]. In the nervous system, alternative autophagy contributes to neuronal proteostasis alongside canonical autophagy, as evidenced by neurodegenerative phenotypes in Wipi3-deficient mice and their phenotypic overlap with Atg7 knockout models [50]. In the immune system, particularly in intestinal mucosal defense, alternative autophagy enables antibacterial responses even in the absence of core canonical components. TRIM31-mediated alternative autophagy in gut epithelial cells facilitates LC3-independent bacterial clearance, offering a potential mechanism to counteract pathogen evasion of canonical autophagy [15]. These findings suggest that alternative autophagy might be not merely a backup system, but a specialized and context-dependent mechanism employed during development, cellular stress, and immune defense (Table 4). As a representative example of such context specificity, increasing attention has been directed toward its role in the heart. The heart depends heavily on mitochondria for continuous energy production, since cardiomyocytes have limited capacity for regeneration and function under high metabolic demand. Even mitochondrial dysfunction can compromise contractile performance and precipitate pathological remodeling. Autophagy provides an essential quality control mechanism by removing damaged mitochondria and preserving cellular homeostasis. Through this process, cardiomyocytes can sustain energy balance, limit oxidative stress, and adapt to hemodynamic or metabolic stressors [69]. Pioneering work from the Dr. Sadoshima group first revealed the involvement of alternative autophagy in cardiac physiology and various stress responses (Figure 4) [13,14,46,47].

### 5.1. Myocardial Infarction

Myocardial infarction (MI) remains a leading cause of death worldwide, primarily caused by the sudden occlusion of coronary arteries, which deprives cardiomyocytes of oxygen and nutrients [71]. Under these conditions, to protect mitochondria from ROS damage, mitochondrial autophagy is generally activated as a protective mechanism. While classical pathways such as Parkin-mediated mitophagy have been widely studied, recent investigations have uncovered that alternative mitophagy takes precedence during ischemic stress [56]. Saito et al. provided compelling evidence that during myocardial ischemia, an alternative mitophagy pathway is activated preferentially over canonical mitophagy [13]. Rab9-positive autophagosomes were observed to engulf damaged mitochondria, and genetic interventions confirmed that this process required Ulk1 and Rab9 rather than Atg5 and Atg7 [5,13,46]. The alternative mitophagy complex, composed of Ulk1, Rab9, Ripk1 and Drp1, was responsible for cardioprotective mitophagy, removing dysfunctional mitochondria to maintain cellular viability during ischemic stress in the heart. The mechanistic cascade is initiated by AMPK activation, which phosphorylates Ulk1, triggering downstream phosphorylation of Rab9 at Ser179 and Drp1 at Ser616, facilitating mitochondrial fission, sequestration and degradation. Phosphorylation-deficient Rab9 S179A knock-in mice suffered greater cardiac injury upon ischemia, despite having intact canonical autophagy [13]. Additionally, inhibition of Golgi-derived membranes with brefeldin A selectively blocks this pathway, highlighting the Golgi as a membrane source unique to alternative autophagy [5]. This reveals that alternative autophagy is not merely a backup, but rather a crucial frontline response in certain stress conditions.

### 5.2. Pressure Overload Condition

In response to sustained hemodynamic stress, such as that induced by transverse aortic constriction (TAC), the heart initially develops a compensatory hypertrophic response [72]. In this condition, canonical LC3-dependent autophagy is transiently and mildly activated, followed by a switch to alternative mitophagy for robust mitochondrial removal. Importantly, Atg7-deficient hearts could still clear mitochondria under prolonged pressure overload condition up to over a week, whereas Ulk1-deficient hearts could not. In this model, cardiomyocyte-specific Ulk1 knockout (Ulk1cKO) mice develop cardiac dysfunction within days, indicating that the alternative mitophagy pathway may be essential for maintaining an effective compensatory hypertrophic response during the early phase of pressure overload [14]. Pharmacological activation Via TAT-Beclin1 enhances both canonical and alternative mitophagy and can reverse cardiac impairment in Ulk1cKO mice, suggesting functional crosstalk and potential therapeutic synergy.

Recent studies have proposed the therapeutic potential of TAT-Beclin1, a synthetic peptide that mimics the functional domain of Beclin1 and promotes autophagy by disrupting the Beclin1-Bcl-2 interaction, thereby releasing Beclin1 to activate autophagy [27]. While TAT-Beclin1 is widely recognized as an inducer of canonical autophagy, emerging evidence suggests that it may also activate alternative autophagy and mitophagy in cardiomyocytes [14]. Notably, in pressure-overloaded hearts, TAT-Beclin1 administration restored mitophagy levels and improved cardiac function. However, these protective effects were abolished in Drp1-hetero-cKO mice and Beclin1-hetero-KO mice, highlighting the essential role of both Drp1 and Beclin1 in mediating TAT-Beclin1-induced mitophagy activation and cardioprotection [59]. Mechanistically, TAT-Beclin1 activates Beclin1, a central regulator of both canonical and alternative autophagy. Thus, its administration can stimulate both pathways. However, the mechanism by which Beclin1 selectively regulates these distinct pathways remains unclear. Our recent identification of TMEM9 as an alternative autophagy-specific Beclin1-binding partner suggests that changes in binding partners of Beclin1 may determine pathway specificity [17]. Understanding how Beclin1 complexes are differentially assembled could enable selective activation of autophagy types, offering a potential strategy to enhance cardiac recovery under pathological stress.

### 5.3. Diabetic Cardiomyopathy

Diabetic cardiomyopathy is characterized by myocardial dysfunction and remodeling that occur independently of hypertension, coronary artery disease, or valvular heart disease [73]. Chronic metabolic stress alters mitochondrial substrate utilization, favoring fatty acid oxidation and increasing oxidative burden [74]. While reports vary regarding autophagy and mitophagy status in diabetic hearts, recent evidence indicates that alternative mitophagy becomes increasingly active during prolonged high-fat diet (HFD) exposure, even as canonical autophagy weakens [46]. Tong et al. demonstrated that during chronic HFD feeding, Rab9 expression is upregulated via TFE3, a transcription factor linked to lysosomal biogenesis and autophagy gene regulation, resulting in sustained activation of Ulk1-Rab9-mediated mitophagy. Notably, enhanced expression of Rab9 can rescue mitophagy levels and prevent cardiac dysfunction even in Atg7-deficient mice, pointing to the compensatory capability of the alternative pathway during diabetic cardiomyopathy [46]. However, the precise regulatory dynamics between TFE3 and its homolog transcription factor EB (TFEB) in coordinating canonical and alternative autophagy remain unresolved.

As aforementioned, a large protein complex consisting of Ulk1, Rab9, Ripk1, and Drp1 is formed to stimulate Drp1 phosphorylation at Ser616, thereby activating alternative mitophagy during energy stress condition in the heart [13]. Building on this, Drp1 has emerged as a key regulator of alternative mitophagy [47]. Interestingly, Drp1 also plays an essential role in canonical mitophagy, where it is likewise activated through phosphorylation and induces mitochondrial fission by generating mechanical force to constrict and divide mitochondria [59,60]. A recent study demonstrated that during the acute phase of high-fat diet exposure, typically around three weeks of feeding, Drp1 interacts with Beclin1 to promote canonical LC3-dependent mitophagy and facilitate the clearance of damaged mitochondria. In contrast, during prolonged high-fat diet exposure lasting up to twenty weeks, Drp1 is phosphorylated at Ser616 and translocates to mitochondria-associated ER membranes, where it interacts with Rab9 to initiate alternative mitophagy, independent of the LC3 conjugation system [47]. These findings suggest that Drp1 serves as a versatile and dynamic regulator of mitophagy, functioning as a molecular switch between canonical and alternative pathways in response to evolving metabolic stress. Maintaining Drp1 activity may therefore be crucial for preserving mitochondrial quality and cardiac function in diabetic and obesity-related cardiomyopathy. However, it is important to note that in most current models, the membrane source for alternative autophagy is primarily thought to be derived from the Golgi apparatus rather than the endoplasmic reticulum [5]. Therefore, the observation that alternative mitophagy may initiate at mitochondria-associated ER membranes warrants further investigation. Clarifying the origin of autophagic membranes in this context will be essential for fully understanding the spatial regulation of alternative mitophagy and its implications in diabetic cardiomyopathy conditions.

Collectively, these insights position alternative mitophagy not as a redundant pathway but as a dynamic, stress-responsive system critical for preserving mitochondrial integrity and cardiac function. Ongoing studies aimed at decoding its regulatory mechanisms and tissue-specific roles may unlock new therapeutic avenues for combating cardiovascular disease.

## 6. Conclusions

Autophagy in mammalian cells operates through two distinct pathways, canonical (Atg5-Atg7-dependent) and Rab9-dependent, Atg5-Atg7-independent alternative autophagy, both critically regulated by the Beclin1 complex [5,24]. While canonical autophagy relies on LC3 conjugation, the alternative pathway engages distinct regulators such as TMEM9, WIPI3, and TRIM31, enabling Rab9-decorated autophagosome formation even when canonical components are compromised [1,15,17,50]. Growing evidence indicates that alternative autophagy is not merely a backup system but a context-dependent mechanism essential for mitochondrial quality control, proteostasis, and host defense [13,14,15]. Among various tissues, the heart represents its physiological importance, where alternative mitophagy protects cardiomyocytes during ischemic stress, pressure overload, and metabolic challenge [13,14,47]. Collectively, these findings emphasize the central role of the Beclin1 complex in flexibly orchestrating both autophagy modes and highlight the therapeutic potential of targeting alternative autophagy in disease.

Despite advances in canonical autophagy research, the molecular mechanisms governing alternative autophagy remain incompletely understood. A major unanswered question in alternative mitophagy is how Rab9 identifies its cargo, particularly damaged mitochondrial membranes. In the canonical pathway, numerous LC3-interacting receptors and adaptors have been identified, whereas in alternative mitophagy the existence and identity of specific receptors/adaptors remain largely unknown. Moreover, LC3 facilitates autophagosome–lysosome fusion via the HOPS complex, but how this fusion event occurs in LC3-independent alternative autophagy is unclear [48]. Progress has also been limited by the absence of pathway-specific detection methods. LC3-dependent canonical autophagy is easily monitored through LC3 lipidation and puncta formation, whereas Rab9-dependent alternative autophagy is difficult to track because Rab9 is also present in late endosomes and other vesicular compartments, limiting its specificity as a marker. Technical innovations that enable specific and quantitative measurement of alternative autophagy activity will be critical for moving the field forward.

From a biomedical perspective, recent studies suggest that alternative autophagy is not simply redundant but a highly specialized, context-dependent system. It holds promise for compensating canonical autophagy defects in neurodegeneration or aging through targeted activation (e.g., Dram1 induction, Bcl-2 inhibition). Conversely, in conditions such as pancreatitis or tumorigenesis, where alternative autophagy may promote pathology, selective inhibition could be beneficial. Understanding how upstream sensors (e.g., AMPK, mTOR, p53, MAPKs) direct autophagic flux toward canonical or alternative routes remain a major challenge [7,44,69,75]. Unraveling these regulatory differences may ultimately enable the development of therapeutic approaches that selectively target canonical versus alternative autophagy.

In conclusion, the Beclin1 complex functions as a pivotal hub linking canonical and alternative autophagic pathways. The ability of cells to reconfigure Beclin1-binding partners, such as TMEM9 in alternative autophagy, may determine pathway choice and specificity. As our understanding of this regulatory architecture deepens and biomarkers like phospho-Rab9 (Ser179) are validated, the potential to selectively manipulate autophagy pathways for therapeutic gain becomes increasingly tangible [13]. The “alternative” is no longer peripheral, it is a critical and adaptive layer of the autophagic network that preserves cellular integrity in health and disease. From a drug discovery perspective, screening for small molecules that specifically activate alternative autophagy (via Ulk1, Dram1, or TMEM9) represents an exciting frontier. Such compounds might bypass canonical autophagy blockades (e.g., Atg5 deficiency or Bcl-2 overexpression) and restore autophagic flux. Conversely, in scenarios such as organ transplantation or acute pancreatitis, dampening alternative autophagy could mitigate tissue damage. The key challenge ahead is achieving pathway-specific modulation, but the unique regulators of alternative autophagy (Rab9, TMEM9, Wipi3, Dram1, TRIM31, etc.) provide promising and more selective molecular targets.

## Figures and Tables

**Figure 1 ijms-26-09151-f001:**
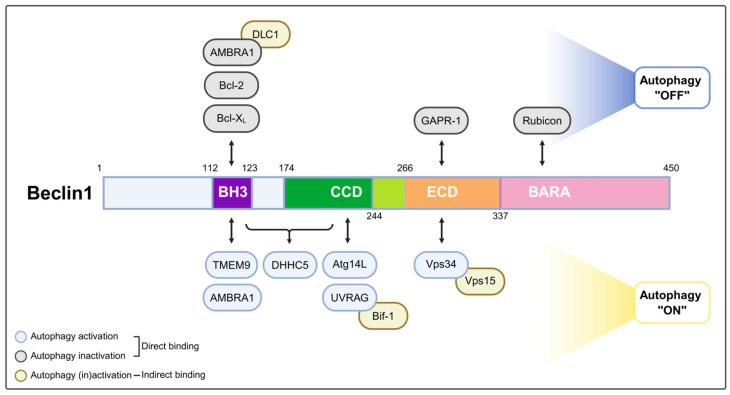
Beclin1 domain map and autophagy regulators. Schematic of human Beclin1 (Beclin1; aa 1–450). Domains are color-coded: BH3/Bcl-2 homology 3 (aa 112–123, purple), CCD/coiled-coil domain (aa 174–244, green), ECD/evolutionarily conserved domain (aa 244–337, orange), and BARA/β-α repeated autophagic-specific domain (aa 337–450, pink). Arrows indicate reported binding regions. Capsule outlines denote function: blue = autophagy activators, gray = inhibitors, gold = indirect regulators (acting via Beclin1-associated partners). Created in BioRender. Baek, S. (2025) https://BioRender.com/6jy9ywr.

**Figure 2 ijms-26-09151-f002:**
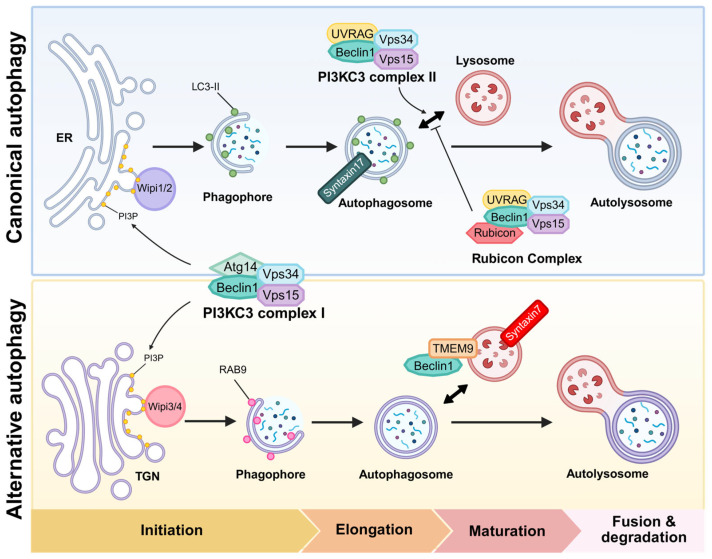
Beclin1 complexes in canonical and alternative autophagy. Beclin1 forms a core component of the PI3KC3 complex, which phosphorylates phosphatidylinositol (PI) to generate PI3P, a lipid essential for autophagosome formation. In canonical autophagy, the PI3KC3 complex II regulates autophagosome maturation, a process that is negatively modulated by Rubicon binding to the complex. In the context of alternative autophagy, Beclin1 also mediates autophagosome maturation through its interaction with the lysosomal membrane protein TMEM9. Syntaxin17 (STX17) is an autophagosome-specific soluble N-ethylmaleimide-sensitive factor attachment protein receptor (SNARE) protein, whereas STX7 not only mediates late endosome-lysosome fusion, but also regulates the fusion step in alternative autophagy. Created in BioRender. Baek, S. (2025) https://BioRender.com/bharqpu.

**Figure 3 ijms-26-09151-f003:**
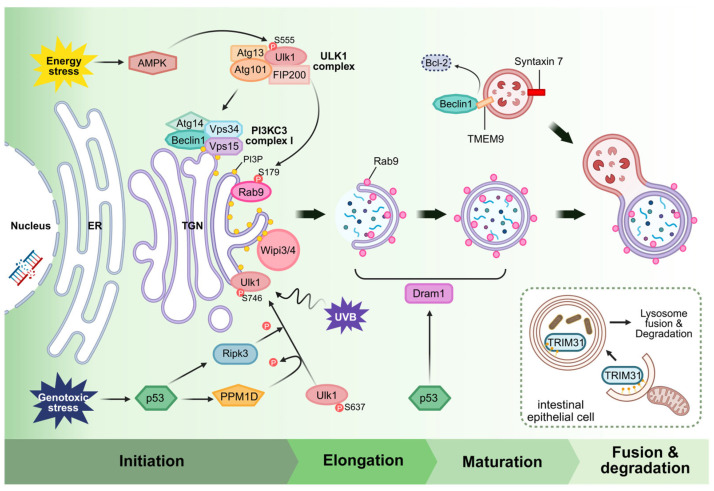
Roles and physiological contexts of Rab9-dependent alternative autophagy. Rab9-positive autophagic membranes. TMEM9, a lysosomal membrane protein, binds to Beclin1 and helps activate the Beclin1-PI3KC3 complex at Rab9-positive membranes, promoting local PI3P production. PI3P subsequently recruits Wipi3 (and possibly Wipi4), a PI3P effector essential for elongation and expansion of the isolation membrane in the alternative autophagy pathway. Dram1, induced by p53 under genotoxic stress, supports the closure and maturation of these autophagosomes in an LC3-independent manner. The mature, Rab9-positive autophagosome subsequently fuses with lysosomes, leading to degradation of cellular cargo. This pathway allows cells to maintain autophagic flux under conditions where the canonical Atg5/Atg7-dependent machinery is impaired. Created in BioRender. Baek, S. (2025) https://BioRender.com/jye26kd.

**Figure 4 ijms-26-09151-f004:**
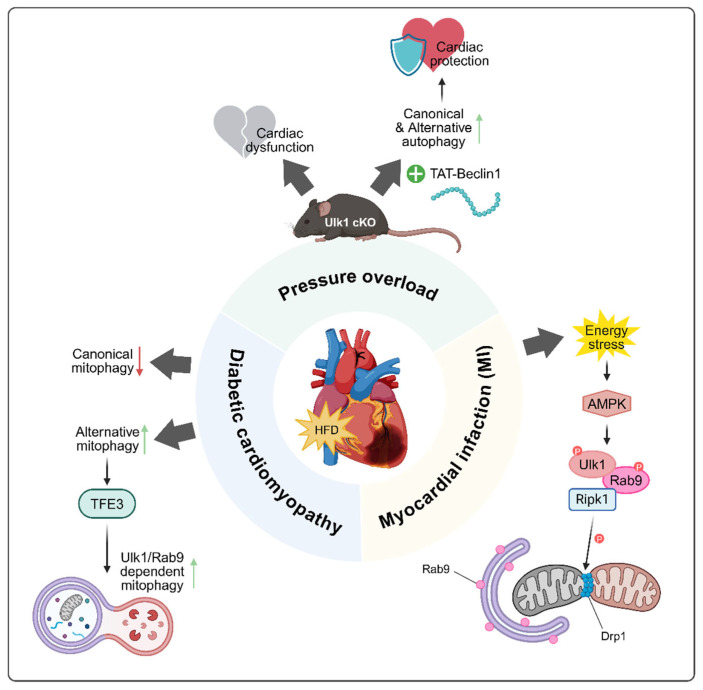
The role of alternative autophagy in cardiac disease mechanisms. During myocardial infarction, Rab9-positive autophagosomes preferentially mediate alternative mitophagy to remove damaged mitochondria in the Ulk1- and Rab9-dependent, but Atg5/7-independent manner. Under pressure overload, mitochondrial clearance shifts from LC3-dependent autophagy to alternative mitophagy, which remains functional in Atg7-deficient hearts and can be enhanced by TAT-Beclin1. In diabetic cardiomyopathy, high-fat diet-induced transcription factor binding to IGHM enhancer 3 (TFE3) activation upregulates Rab9, sustaining Ulk1-Rab9-mediated mitophagy as canonical autophagy declines. Drp1 acts as a dynamic regulator, switching between canonical and alternative mitophagy in response to metabolic stress. Created in BioRender. Baek, S. (2025) https://BioRender.com/sxezde8.

**Table 1 ijms-26-09151-t001:** Beclin1 domains, interacting partners, and their roles in autophagy regulation.

Beclin1 Domain	Protein	Direct/Indirect	Function	Reference
BH3	Bcl-2 or Bcl-X_L_(Bcl2L1)	Direct	Suppresses autophagy under normal conditions; stress or competition allows Beclin1-Vps34 complex formation	[18,23,24]
	TMEM9	Direct	Binds directly to Beclin1 and activates the Beclin1-Vps34 complex in alternative autophagy	[17]
	Autophagy and Beclin1 regulator 1 (AMBRA1)	Direct	Promotes Beclin1-Vps34 complex formation during starvation; enhances endoplasmic reticulum (ER) recruitment and autophagosome nucleation	[25]
CCD	Atg14L	Direct	Recruits the Beclin1-Vps34-Vps15 complex to ER membranes, enabling PI3P production and initiating phagophore formation	[21,22,26]
	UVRAG	Direct	Enhances endosomal trafficking, autophagosome maturation, and lysosome fusion through interaction with PI3KC3 complex II	[22]
ECD and BARA	Vps34	Direct	Produces PI3P, a crucial lipid signal for autophagy initiation	[20]
	Rubicon	Direct	Inhibits autophagosome maturation by forming a complex with Beclin1-UVRAG-Vps34-Vps15	[21,26]
	Golgi-associated plant pathogenesis-related protein 1 (GAPR1; GLIPR2)	Direct	Inhibits autophagy by preventing Beclin1 complex assembly at lysosomal or Golgi membranes	[27]
Other interaction	Vps15 (p150)	Indirect (via Vps34)	Regulates Vps34 and promotes PI3P synthesis during autophagy initiation	[20]
	Dynein light chain 1 (DLC1)	Indirect (via AMBRA1)	Anchors the AMBRA1-Beclin1-Vps34 complex to dynein under nutrient-rich conditions; releases upon Ulk1-mediated phosphorylation during starvation	[25]
	Bax-interacting factor 1 (Bif-1)	Indirect (via UVRAG)	Promotes autophagy by interacting with Beclin1 and UVRAG to activate the PI3KC3 complex, facilitating autophagosome formation	[28]
	DHHC-type palmitoyltransferase 5 (DHHC5)	BH3, CCD	Promotes the S-palmitoylation of Beclin1, enhancing the formation of the Atg14L-containing PI3KC3 complex critical for autophagy initiation	[29]

**Table 2 ijms-26-09151-t002:** Post-translational modifications of Beclin1 and their functional consequences.

PTMs of Beclin1	Site of PTMs	Kinases/Enzymes	Function	Reference
Phosphorylation	Ser14	Ulk1	Activation of the Vps34 complex kinase	[35]
Phosphorylation	Ser91 and Ser94	AMPK	Promotion of autophagy	[34]
Phosphorylation	Thr108	Mst1	Promotion of the interaction between Beclin-1 and Bcl-2/Bcl-X_L_	[37]
Phosphorylation	Thr119	DAPK	Disruption of the association with the anti-apoptotic protein Bcl-X_L_	[33]
Phosphorylation	Ser234 and Ser295	Akt	Suppression of autophagy	[36]
S-palmitoylation	Cys137 and Cys159	DHHC5	Localization and autophagy initiation of Beclin1	[29]

**Table 3 ijms-26-09151-t003:** Key molecular and functional distinctions between canonical and alternative autophagy.

Characteristics	Canonical Autophagy	Alternative Autophagy	Reference
Primary stimuli	Nutrient deprivation, general stress	DNA damage, ischemia, prolonged pressure overload, prolonged metabolic stress	C—[4,6,7,8]
			A—[13,14,44,45,46,47]
Phosphorylation sites of activated Ulk1	At Ser317/Ser637/Ser777 by AMPK	At Ser746 by receptor-interacting serine-threonine kinase 3 (Ripk3)	C—[4,7]
			A—[13,44]
Phosphorylation sites of inactivated Ulk1	At Ser638/Ser758 by mTORC1	-	C—[6]
Key regulators	Atg5, Atg7, LC3, Wipi1/2, PI3KC3 complex I/II, STX17	Rab9, TMEM9, damage-regulated autophagy modulator 1 (Dram1), STX7, Wipi3/4	C—[1,2,3,10,21,23,39,41,48]
			A—[5,13,14,17,40,45,49,50,51,52]
Biomarker	Atg5, Atg7, LC3	Rab9, TMEM9, Wipi3, Dram1	C—[1,2,3]
			A—[5,13,14,17,45,50]
Membrane source	ER-derived membranes	TGN-derived membranes	C—[10,43]
			A—[5,13,49,53]
Components of Beclin1 complex	Beclin1-Vps34-Vps15 complex; regulated by Atg14L, UVRAG, Rubicon, AMBRA1	Beclin1-TMEM9	C—[19,20,21,22,25,26]
			A—[17]
Membrane elongation/maturation	Atg12-Atg5-Atg16L1 complex, LC3 conjugation system	Rab9, Dram1, Wipi3	C—[1,2,3,12]
			A—[5,45,50]
Fusion machinery	STX17-synaptosomal-associated protein 29 (SNAP29)-vesicle-associated membrane protein 8 (VAMP8) with homotypic fusion and protein sorting (HOPS), pleckstrin homology domain-containing family M member 1 (PLEKHM1)/ectopic P-granules autophagy protein 5 (EPG5)	STX7	C—[48,54,55]
			A—[5,50]
Autophagosome marker	LC3 and gamma-aminobutyric acid receptor-associated protein (GABARAP) family	Rab9	C—[1]
			A—[5,13,14]
Cargo substrates	Bulk and selective substrates (mitochondria, etc.)	Mitochondria and intestinal bacteria	C—[1,2,3,56]
			A—[13,14,15,46,57,58]
Function	Homeostasis, adaptation to starvation and disease	Cardioprotection in heart disease, neuronal proteostasis, mitochondrial clearance and antibacterial defense	C—[1,2,3,42,43]
			A—[13,14,15,46,47,50,57,59,60]

Abbreviations: C—canonical autophagy; A—alternative autophagy.

**Table 4 ijms-26-09151-t004:** Pathophysiological roles of alternative autophagy across organs and disorders.

Organ	Disease/Condition	Pathophysiological Role	Reference
Heart	Myocardial infarction (MI)	Protective	[13]
	Pressure overload (TAC)		[14]
	Diabetic cardiomyopathy		[47]
Brain	Neurodegenerative disorders	Protective	[50]
Intestine	Crohn’s disease	Protective	[15]
Pancreas	Acute pancreatitis	Detrimental	[70]
Skin	Sunburn	Protective	[68]
	UVB-induced cutaneous inflammation		

## Data Availability

Not applicable.

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
