# Peer review of "The Role of the Beclin1 Complex in Rab9-Dependent Alternative Autophagy"

_ijms, 2025, doi:10.3390/ijms26189151_

Round 1

Reviewer 1 Report

Comments and Suggestions for Authors

The manuscript by Sohyeon Baek and co-authors examines the alternative autophagy pathway and in specific the role of the Beclin1 complex in Rab9-dependent alternative autophagy. Direct comparisons are made to the canonical autophagy pathway, with common and different mechanisms being designated. In addition, the contribution of alternative autophagy is described in physiological and pathophysiological settings. This is a well-written review manuscript but would benefit from some revisions and condensation of text.  

1) How is alternative autophagy induced? Some stress factors are mentioned in the conclusion but would useful to the reader to be clarified at earlier points. Similarly, what determines whether the canonical or alternative autophagy pathway will be activated?

2) The information on TMEM9 that is included on page 11, lines 414-426, has already been described in the previous two paragraphs. To avoid unnecessary duplications, please include any required details in the previous paragraphs. This would aid in a more focused and condensed text.  

3) Along those lines, please try to condense the first three paragraphs of the conclusion section. These are a summary of the key points of the main text rather than conclusions.

4) Revision of the title of section-5 to ‘Physiological and Pathophysiological Roles of Alternative Autophagy’ may be more appropriate since the first paragraph of this section is focused on physiological aspects and not disease.   

5) Multiple alternative autophagy examples involve mitochondrial clearance. Is the alternative autophagy pathway specific for this organelle or are there other examples of organelles/proteins that are degraded through alternative autophagy?

6) In section 5, the authors discuss the impact of alternative autophagy in pathophysiology. This section is specifically focused in the heart but the rational behind this is not clear. Has type this of autophagy been more widely studied in the heart and therefore there is more knowledge on this, is this mechanism more important in the heart or are there other reasons behind the focus in the heart. A clarification on this would be useful as throughout the text other diseases (neurodegeneration, aging or cancer) are also mentioned but not explored in detail.   

7) The authors can consider the possibility of a figure depicting the post-translational modifications of Beclin, the enzymes/kinases mediating this and the consequent effect on autophagy.  

8) In text citations, some instances are in italics or in bold. Please correct for consistency purposes.

Author Response

Comments 1: How is alternative autophagy induced? Some stress factors are mentioned in the conclusion but would useful to the reader to be clarified at earlier points. Similarly, what determines whether the canonical or alternative autophagy pathway will be activated?

Response 1: We appreciate the reviewer’s insightful comment. To address this, we have added a new Table 2 summarizing the distinct differences between canonical and alternative autophagy, including their key stimuli, which we believe will be helpful for readers.

Regarding the factors determining canonical versus alternative autophagy activation, our previous work (Nah et al., 2020) demonstrated that canonical autophagy is rapidly induced in response to acute pressure overload, whereas the Ulk1-dependent alternative pathway remains chronically sustained for up to one week under the same conditions. These findings suggest that canonical and alternative autophagy are differentially engaged depending on the nature and duration of the stress stimulus. Canonical autophagy likely mediates the immediate adaptive response, whereas alternative autophagy contributes to longer-term or chronic stress adaptation. We have revised the manuscript to clarify these points and highlight the timing- and stress-specific regulation of canonical versus alternative autophagy (lines 541–548 and 269–271).

Comments 2: The information on TMEM9 that is included on page 11, lines 414-426, has already been described in the previous two paragraphs. To avoid unnecessary duplications, please include any required details in the previous paragraphs. This would aid in a more focused and condensed text.  

Response 2: Thank you for this helpful comment. We have carefully revised the section and removed the redundant discussion of TMEM9. The key details originally presented in lines 397–400 have been integrated into the earlier paragraphs where TMEM9 was first introduced.

Comments 3: Along those lines, please try to condense the first three paragraphs of the conclusion section. These are a summary of the key points of the main text rather than conclusions.

Response 3: We thank the reviewer for this helpful comment. As suggested, in the revised manuscript, we have condensed and restructured the conclusion section. Rather than reiterating detailed content from the main text, this section now emphasizes the central message, including key unanswered mechanistic questions (lines 627–634), emerging therapeutic opportunities targeting alternative autophagy (lines 652-656), and technical advances needed to better study alternative autophagy (lines 634-640). Collectively, these revisions make the conclusion more concise and focused.

Comments 4: Revision of the title of section-5 to ‘Physiological and Pathophysiological Roles of Alternative Autophagy’ may be more appropriate since the first paragraph of this section is focused on physiological aspects and not disease.   

Response 4: We appreciate this helpful suggestion. The title of section 5 has been revised to “Physiological and Pathophysiological Roles of Alternative Autophagy,” which more accurately reflects the scope of the section.

Comments 5: Multiple alternative autophagy examples involve mitochondrial clearance. Is the alternative autophagy pathway specific for this organelle or are there other examples of organelles/proteins that are degraded through alternative autophagy?

Response 5: Thank you for this important question. To date, most studies have primarily reported the role of alternative autophagy in mitophagy, and evidence for its involvement in other forms of selective autophagy remains limited. However, there are a few reports describing LC3-independent autophagy, such as TRIM31-mediated xenophagy, which degrades intracellular pathogens (lines 433-445). This may reflect the fact that mitophagy has been the most actively investigated form of selective autophagy to date. We have briefly added this point to the revised manuscript (lines 447-453 and lines 466-467).

Comments 6: In section 5, the authors discuss the impact of alternative autophagy in pathophysiology. This section is specifically focused in the heart but the rational behind this is not clear. Has type this of autophagy been more widely studied in the heart and therefore there is more knowledge on this, is this mechanism more important in the heart or are there other reasons behind the focus in the heart. A clarification on this would be useful as throughout the text other diseases (neurodegeneration, aging or cancer) are also mentioned but not explored in detail.   

Response 6: We thank the reviewer for raising this important point. In the revised manuscript, we have clarified our rationale for focusing Section 5 on the heart. Alternative autophagy has indeed been more extensively characterized in the heart than in other organs, resulting in a stronger body of mechanistic evidence. Moreover, cardiac physiology is highly dependent on mitochondrial quality control, and alternative autophagy appears to be particularly important for maintaining cardiac homeostasis under stress conditions.

While we briefly mention its roles in somatic cell reprogramming, neurodegeneration, and the intestinal immune system, the available data in these contexts remain relatively limited. To better inform readers, we now explicitly state in lines 494–501 that the heart has been the most widely studied organ for alternative autophagy. In addition, we have included a new Table 4 summarizing the pathophysiological roles of alternative autophagy across different tissues and disease settings to provide a broader perspective.

Comments 7: The authors can consider the possibility of a figure depicting the post-translational modifications of Beclin, the enzymes/kinases mediating this and the consequent effect on autophagy.  

Response 7: We thank the reviewer for this constructive suggestion. In the revised manuscript, we have included a new table (Table 3) summarizing the post-translational modifications (PTMs) of Beclin1, the kinases/enzymes responsible, and their functional consequences on autophagy. This addition provides a clearer overview of how Beclin1 is regulated at the post-translational level, complementing the text description and facilitating better understanding for the reader.

Comments 8: In text citations, some instances are in italics or in bold. Please correct for consistency purposes.

Response 8: We thank the reviewer for pointing out this formatting issue. We have carefully reviewed all in-text citations and corrected them to ensure a consistent style throughout the manuscript.

Reviewer 2 Report

Comments and Suggestions for Authors

This review covers an important and timely topic: the role of the Beclin1 complex in Rab9-dependent alternative autophagy. The authors provide a thorough overview of canonical autophagy and emphasize emerging evidence for Atg5/Atg7-independent pathways. The discussion of Beclin1 as a central scaffold, along with its interaction with novel regulators such as TMEM9, is original and offers translational relevance. Overall, the manuscript is clearly written, well-organized, and will be of interest to researchers in autophagy and cardiovascular biology. That said, several aspects could be further strengthened:

1.Therapeutic Implications: Although potential therapeutic applications are mentioned, the review would benefit from concrete examples, such as small molecules, genetic tools, or clinical contexts. Expanding this section would enhance the translational impact.

2.Although the manuscript discusses recent advances on TMEM9 and Rab9-dependent autophagy, several important studies from the past 3–5 years may have been overlooked.

Author Response

Comments 1: Therapeutic Implications: Although potential therapeutic applications are mentioned, the review would benefit from concrete examples, such as small molecules, genetic tools, or clinical contexts. Expanding this section would enhance the translational impact.

Response 1: We thank the reviewer for highlighting this important point. As noted, we briefly discussed potential therapeutic applications, including Tat-Beclin1, which may activate both canonical and alternative autophagy and thereby improve cardiac function in response to pressure overload (line 548-550 and lines 560-567). However, the current evidence remains limited, and additional mechanistic and translational studies are strongly needed.

In response to the reviewer’s suggestion, we have added a new table (Table 2) summarizing key distinctions between canonical and alternative autophagy, including known activators and genetic tools. This revision strengthens the translational perspective of the review.

Comments 2: Although the manuscript discusses recent advances on TMEM9 and Rab9-dependent autophagy, several important studies from the past 3–5 years may have been overlooked.

Response 2: We appreciate the reviewer’s insightful suggestion to reassess recent progress in the field. As noted, several important studies have recently advanced our understanding of alternative autophagy, also referred to as Golgi membrane-associated degradation (GOMED). While our review manuscript primarily focuses on the role of the Beclin1 complex in alternative autophagy, rather than comprehensively detailing the pathway as a whole, we have revised the manuscript to acknowledge and incorporate select key molecular and pathophysiological findings from recent literature. These updates are now included in the revised version (lines 447–453, 466–467). Additionally, we have reinforced the manuscript by introducing two new summary tables (Table 2 and Table 4) that highlight emerging mechanistic insights and physiological relevance of alternative autophagy/GOMED compared to canonical autophagy. We thank the reviewer once again for the valuable feedback, which helped improve the scope and relevance of the manuscript.

Reviewer 3 Report

Comments and Suggestions for Authors

The authors provide a comprehensive update on the role of the Beclin1 complex in Rab9-dependent alternative autophagy. This review manuscript demonstrates good organization and clarity in presentation. To enhance the scientific impact of the manuscript and its contribution to the field, the reviewer recommends addressing the following points:

1. Mechanistic Distinction and Comparative Analysis: The manuscript would benefit from a more detailed comparative analysis between canonical and alternative autophagy pathways. Please include: (a) A comprehensive table or figure comparing key molecular components, regulatory mechanisms, and functional outcomes of both pathways, (b) Discussion of the cellular conditions that favor one pathway over the other, and (c) Analysis of potential crosstalk or coordination between these two autophagy systems.

2. Clinical and Pathophysiological Relevance: The significance of Beclin1 complex and Rab9-dependent alternative autophagy in human health and disease requires substantial expansion. The authors should address: (a) Specific human diseases where this pathway is dysregulated or therapeutically relevant, (b) Comparative analysis with experimental disease models beyond cardiac conditions, (c) Potential biomarker applications, and (d) Discussion of how dysfunction in this pathway contributes to disease pathogenesis across different organ systems.

3. Future Perspectives and Research Directions: Please elaborate on future research perspectives by discussing: (a) Key unanswered mechanistic questions in the field, (b) Emerging therapeutic opportunities targeting this pathway, (c) Technical advances needed to better study alternative autophagy in vivo, and (d) Potential applications in precision medicine approaches.

4. Reference Formatting: There are inconsistencies in reference formatting throughout the manuscript. Please carefully review and standardize the citation style according to journal guidelines, ensuring consistency in author names, journal abbreviations, and formatting structure.

Author Response

Comments 1: Mechanistic Distinction and Comparative Analysis: The manuscript would benefit from a more detailed comparative analysis between canonical and alternative autophagy pathways. Please include: (a) A comprehensive table or figure comparing key molecular components, regulatory mechanisms, and functional outcomes of both pathways, (b) Discussion of the cellular conditions that favor one pathway over the other, and (c) Analysis of potential crosstalk or coordination between these two autophagy systems.

Response 1: We thank the reviewer for this valuable suggestion. In the revised manuscript, we have:
(a) included a new comparative table (Table 2) that systematically summarizes the key molecular components, regulatory mechanisms, and functional outcomes distinguishing canonical and alternative autophagy.

(b) Our previous study (Nah et al., 2020) provided evidence that canonical autophagy is rapidly activated in response to acute pressure overload, whereas the Ulk1-dependent alternative pathway remains chronically sustained for up to one week under the same condition. These findings together suggest that canonical and alternative autophagy may be differentially activated depending on the nature and duration of the stress stimulus, canonical autophagy likely mediates the immediate response, whereas alternative autophagy may play a key role in prolonged or chronic stress. We have revised the manuscript to clarify these points and to address the timing and stress-specific regulation of canonical versus alternative autophagy (lines 541–548, and lines 269–271, and new Table 2 and Table 4).

(C) Although canonical and alternative autophagy pathways are generally regarded as independent, they share certain proteins (e.g., Beclin1, Ulk1) while also involving distinct regulators (e.g., TMEM9, Rab9, LC3). At present, the detailed mechanisms of alternative autophagy remain incompletely understood, which limits our ability to comprehensively analyze potential crosstalk between the two systems. However, we have added Table 2 to facilitate a clearer understanding of the molecular and functional distinctions, thereby helping readers better understand crosstalk between canonical and alternative autophagy.

Comments 2: Clinical and Pathophysiological Relevance: The significance of Beclin1 complex and Rab9-dependent alternative autophagy in human health and disease requires substantial expansion. The authors should address: (a) Specific human diseases where this pathway is dysregulated or therapeutically relevant, (b) Comparative analysis with experimental disease models beyond cardiac conditions, (c) Potential biomarker applications, and (d) Discussion of how dysfunction in this pathway contributes to disease pathogenesis across different organ systems.

Response 2: We thank the reviewer for pointing this out.

(a) We added a new table (Table 4) summarizing specific human diseases, such as neurodegenerative disorders, metabolic syndromes, and cancers, where alternative autophagy has been implicated. This includes references to recent mechanistic studies and patient-derived data, where available.

(b) While cardiac ischemia has been a primary focus of alternative autophagy research, we agree that broader disease contexts are required. In the revised manuscript, we now highlight its roles in neurodegeneration, cancer, and infection models. For example, in neuronal systems, upregulation of alternative autophagy has been shown to compensate for deficiencies in canonical autophagy, preserving cellular homeostasis (lines 320-336). In cancer, particularly in tumors with elevated Wnt signaling, TMEM9-driven activation of the Rab9-dependent pathway facilitates stress adaptation and survival (lines 398-424). These examples highlight that the relevance of alternative autophagy extends well beyond cardiac physiology. The expanded content has been summarized in a newly added Table 4 for clarity and comparative insight.

(c) The unique regulators of alternative autophagy, such as Rab9, TMEM9, WIPI3, and DRAM1, provide potential biomarkers for disease diagnosis or prognosis. For example, elevated Rab9 activity or TMEM9 upregulation in tumors could serve as indicators of altered autophagy flux that supports cancer cell survival, highlighting their translational potential as biomarkers. We have integrated these points into new Table 2, which outlines candidate biomarkers alongside their associated disease contexts and potential clinical applications.

(d) We also included this concerns in the new table ( 4).

Comments 3: Future Perspectives and Research Directions: Please elaborate on future research perspectives by discussing: (a) Key unanswered mechanistic questions in the field, (b) Emerging therapeutic opportunities targeting this pathway, (c) Technical advances needed to better study alternative autophagy in vivo, and (d) Potential applications in precision medicine approaches.

Response 3: We thank the reviewer for highlighting these critical points regarding the future directions of the field. As suggested, in the revised manuscript we have condensed and restructured the conclusion section. Rather than reiterating detailed content from the main text, this section now emphasizes the central message, including key unanswered mechanistic questions (lines 627–634), emerging therapeutic opportunities targeting alternative autophagy (lines 652-656), and technical advances needed to better study alternative autophagy (lines 634-640). Regarding potential applications in precision medicine, we were unable to provide a meaningful discussion as no studies to date have specifically addressed this aspect of alternative autophagy. Collectively, these revisions make the conclusion more concise and focused.

Comments 4: Reference Formatting: There are inconsistencies in reference formatting throughout the manuscript. Please carefully review and standardize the citation style according to journal guidelines, ensuring consistency in author names, journal abbreviations, and formatting structure.

Response 4: We thank the reviewer for pointing this out. We have carefully reviewed all references in the manuscript and revised them to ensure full consistency with the journal’s citation guidelines. Author names, journal abbreviations, and overall formatting structure have been standardized throughout.